# Between Sand Dunes and Hamadas: Environmental Sustainability of the Thar Desert, West India

**Jiri Chlachula** [1,2]

1  Institute of Geoecology and Geoinformation, Adam Mickiewicz University, 61-680 Poznań, Poland; paleo@amu.edu.pl
2  Environmental Research Centre, 686 03 Stare Mesto, Czech Republic; Altay@seznam.cz

**Abstract:** Extensive geographic areas of the world show a long-term atmospheric moisture deficit. Desertification of Rajasthan is concurrent with the strengthened weather extremality and mean annual air temperature (MAAT) rise over the western part of the Indian subcontinent. The present landscape aridification due to the precipitation decrease and reinforced windiness generates surface-cover dryness, aeolian erosion with a mass sediment transfer, salinity of excessively irrigated lands and groundwater depletion; altogether these pose major geo-environmental threats and settlement risks of the expanding Thar Desert. Livestock-overgrazing of sparse-vegetation contributes to ecological pressure to the fragile wasteland ecosystems with approximately three-quarters of the countryside affected to a certain extent by degradation and >50% exposed to wind erosion. Sand dune stabilisation by the drought-adapted tree plantation, the regional hydrology network regulation and the arid-land farming based on new xerophytic cultigens are the key land-use and mitigation strategies. Specific geomorphic palaeosettings predetermined patterned adaptive forms of the ancient desert inhabitation. Geo- and eco-tourism contributes to the arid-zone socioeconomic sustainability with regard to the rich natural and cultural heritage of the area. This study outlines the main effects of the current climate variations on the pristine and occupied lands of western Rajasthan, and the past and present relief transformations, and reviews the modern anthropogenic responses to desertification.

**Keywords:** Rajasthan; climate change; aridification; sand dunes; environmental impact; landscape degradation; settlement sustainability; geoheritage





## 1. Introduction

Present aridification is a worldwide phenomenon threatening regional ecosystems and economies [1–5]. The trends of globally progressing landscape dryness reflect the changes in atmospheric circulation patterns, solar insolation and windiness contributing to regional climate continentality. The combination of these factors predetermines geomorphic landscape dynamics, groundwater resources and environmental sustainability.

The Thar Desert is the driest place of the arid zone of the NW Indian subcontinent (Figure 1). Past geomorphic processes related to orogenesis and climate change acting on this territory generated a diversity of landforms depending on the structural geology and bedrock intensity weathering and the resulting depositional bodies. Natural landscapes have been changed by modern human activities modifying the original relief and contributing to the present environmental vulnerability and a loss of natural resilience.

The ongoing desertification manifested by an increase of MAAT and a precipitation decrease over most of NW India [6] triggers a mass, large-scale sediment transfer and ground surface salinity (both severely affecting agrarian lands). These natural processes pose major threats to the local ecosystems and limitations to the rural occupation habitats. In western Rajasthan, the annual rainfall is estimated to have dropped by ≈ 15% over the past 100 years [7]. The sparse desert vegetation cover contributes to intensified windiness and a large-scale sand transport generating an active dune-fields' formation.

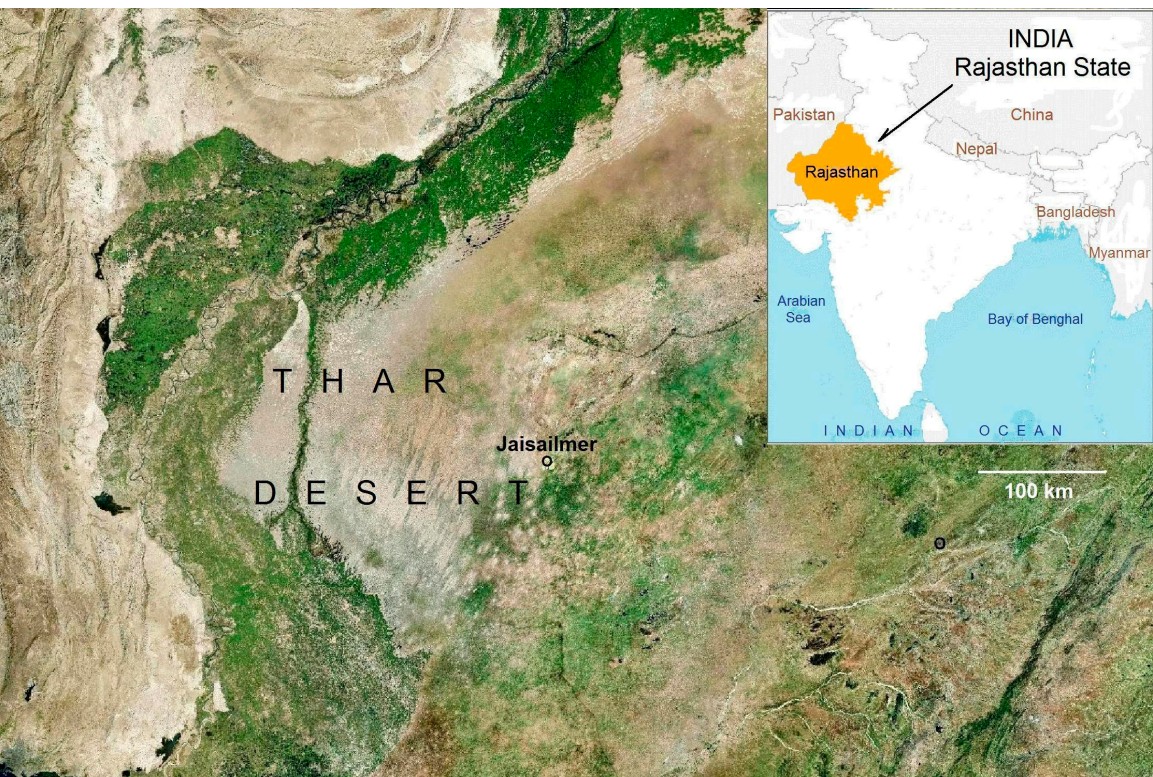

**Figure 1.** Geography of the Thar Desert (West India–East Pakistan) (the satellite map by Google Earth).

The arid zone of NW India shows a differential degree of landscape degradation in terms of environmental stability, natural restoration capability and socioeconomic sustainability because of the variably co-acting adverse natural and human factors. These have a direct bearing on the inhabitability of the traditional settlement loci in the desert. The same concerns the climatically more favourable and densely occupied northern and eastern areas of Rajasthan receiving an increased annual rainfall (300–500 mm) which allows for an intensive, largely irrigation-based agricultural economy. Presently, >5800 km$^2$ of the land are severely degraded, 24,430 km$^2$ degraded, 73,740 km$^2$ moderately degraded, and 526,900 km$^2$ are slightly degraded [8]. The unfavourable landscape shifts are countermeasured by the regional land-management strategies that are aimed at securing the ecological stability of the state. The responding environmental sustainability management includes, among other, an artificial landscape re-vegetation by planting selected xerothermic bushes and trees, fixing the activated sand dune fields; implementation of a diversified land-use policy, mainly promoting adaptive drought-resistant cultigens; and the construction of the rainfall-water retention dams for year-round water availability. These ecology precaution measures against the progressing regional aridification and the large-scale desert sand transfer are intended to secure future industrial growth along with the long-term rising demography particularly due to urbanisation [9].

Rajasthan, as the largest and most arid state of India (342,239 km$^2$), experiences the major social and economic challenges backed-up by increasing revenues from different sectors (agriculture, mining, services, transportation, tourism), reflecting the improving livelihood conditions of the residents. The expanding transport network has facilitated better accessibility to the marginal pristine desert areas for development and trade, as well as new settlements that have moved from the densely populated eastern parts of the state. This progress, in turn, generates serious environmental threats due to the rising population (from ≈1 million to 30 million over the past 50 years, with the current ≈130 people/km$^2$ population density) in the formerly sparsely populated western wasteland area. This leads to a gradual depletion of natural resources (mainly ground water and arable land). At present, about 80% of western Rajasthan is affected by a certain form of land degradation;

>73% is subjected to wind erosion and sand dune deposition [10] (Figures 2 and 3). Vegetation retreat, soil and water erosion, topsoil salinity/alkalinity and industrial (mining and construction) activities represent the other major regional ecological risks.

Aeolian erosion hazards generated by the long-term aridity and the exposed-terrain properties (such as surface roughness, soil moisture balance and the character of vegetation cover as the principal controlling factors) [11], along with the landscape dryness, are the most acute environmental sustainability issues of West India's arid zone regions. These factors acting alongside the local agricultural peculiarities that have been adjusted to the existing topographic and climatic conditions (i.e., soil, groundwater, crop yields, demographic and agroeconomic aspects) define the specific agroecological regions of western Rajasthan [12], which are exposed to landscape degradation to varying extents.

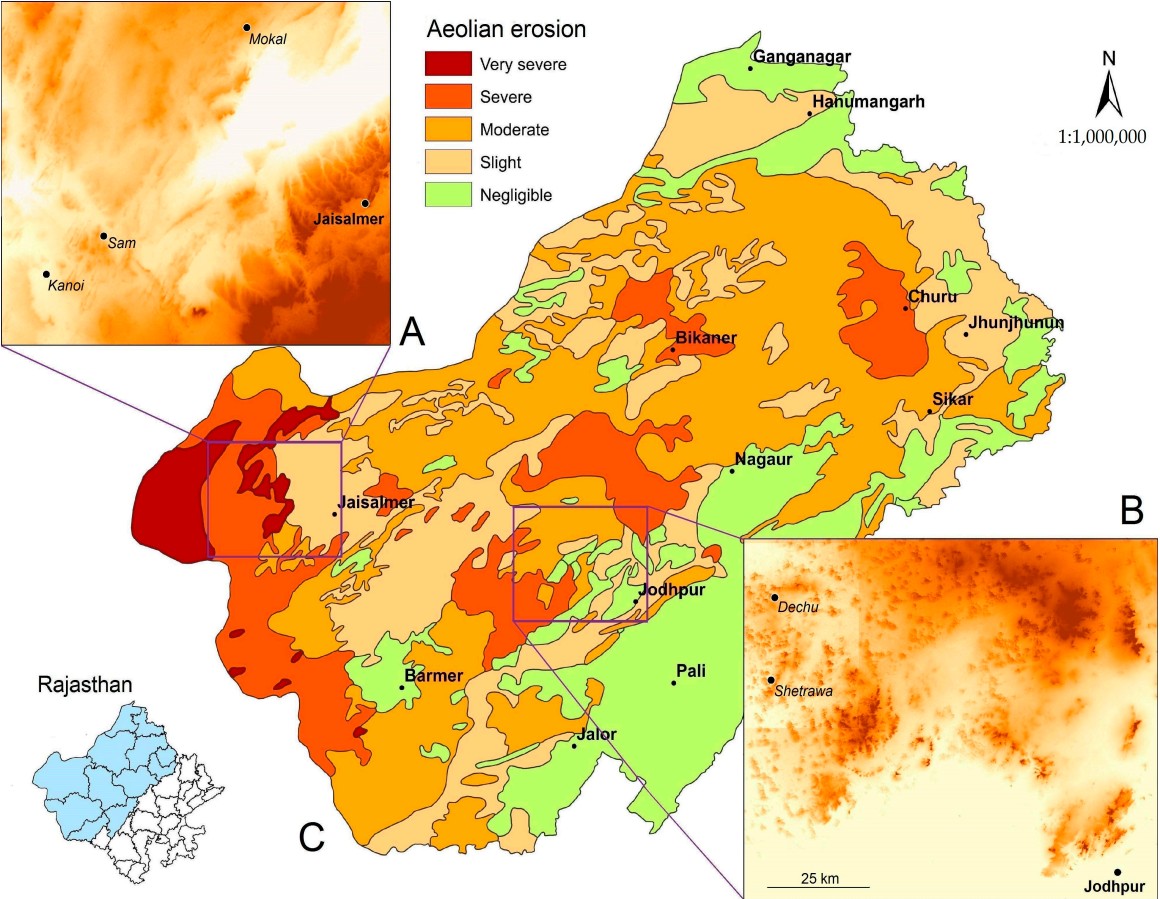

**Figure 2.** Aeolian erosion of western Rajasthan with the Digital Elevation Model (DEM) relief maps of the discussed areas: (**A**) Jaisalmer District and (**B**) Jodhpur District. The western and central parts of the country represent the most extreme SW wind- stream-affected geoenvironment with mass sand sediment-generating processes and the formation of barchanoid and longitudinal sand dune fields. Parabolic and transverse dunes are characterised by less severe, moderate to slight erosion–accumulation settings, respectively. The DEM sector relief (**A**,**B**) processing and projection this study; (**C**) the background wind-erosion map [13].

This paper discusses some of the main issues of the current environmental change in western Rajasthan, its feedback to the desert settlements, and outlines the principal natural and anthropogenic factors towards the present occupation sustainability within the extreme Thar Desert settings. The aim of the study was to provide a comprehensive assessment of the contemporary geoecology and the human geography of the arid zone of NW India in the frame of the territorial management of the dynamically transforming landscapes under the progressing terrestrial aridity and the socioeconomic pressure.

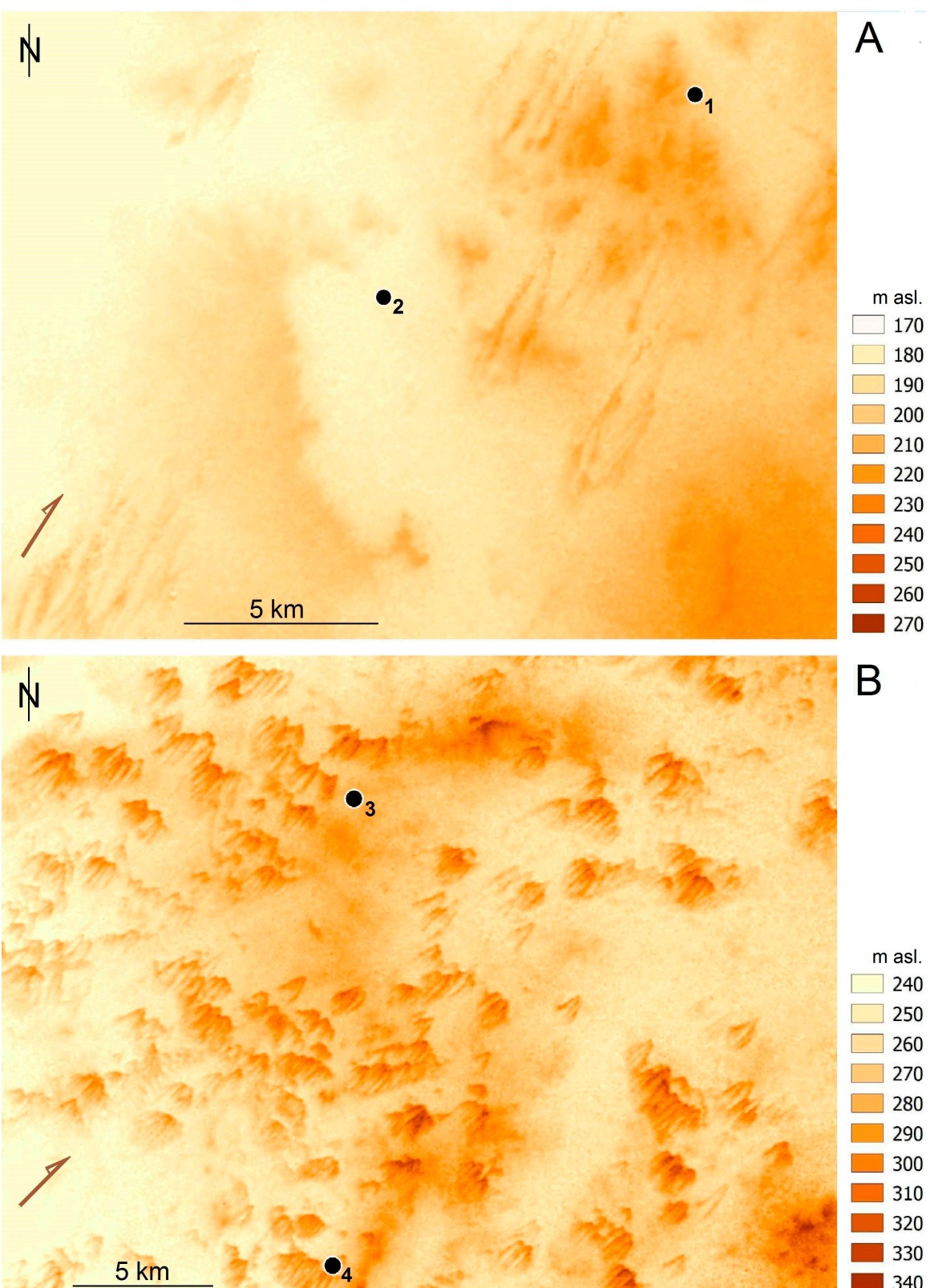

**Figure 3.** (**A**): Digital elevation model/DEM (the relief data SRTM 1 Arc-Second Global, resolution 30 m) with the projection of the longitudinal sand dune fields between Sam (1) and Kanoi (2), the Jaisalmer District. (**B**): DEM of parabolic dunes in the Dechu (3) and Shetrawa (4) area, the Jodhpur District, the western Thar Desert, showing the dominant SW–NE-oriented wind direction and aeolian sediment transport. For the geography of the aerial DEM relief maps see Figure 2.

## 2. Study Area

The Thar Desert (the Great Indian Desert), located in the centre of the hot arid zone of NW India, is one of the major deserts in the subtropical belt (196,000 km$^2$) (Figure 1). Most of the desert, geographically delimited by the Aravalli Hills in the east and the Indus River in the west, belongs to India (85%) and a lesser part to Pakistan (15%). In India, the dry wastelands occupy most of Rajasthan (60%); the remaining portions expand into the neighbouring Haryana and Gurajat states in the north and south, respectively. Very high aridity, along with the seasonal precipitation regime of the eastern (Indian Ocean) monsoons (June–September), predetermines the controlling continental climate. The mean annual precipitation amounts 150–500 mm depending on the relief and the geographical area. The western most arid region receives only ≈130 mm rainfall [14]. Increased wind velocity, with maxima during summer months (June–August) (average 5–6 km/h) and minima (average 2–3 km/h) during winter months (December–February), correlates with the seasonal rising and declining temperature fluctuations, respectively [15].

The average MAAT is ~25 °C with high summer temperatures (33–40 °C) and reduced winter temperatures 9–20 °C [16]. The broad seasonal air temperature range (−5 °C/+50 °C) contributes to an intensified physical and chemical sediment weathering. The increased windiness (with a wind speed >3 m/s) generates a large-scale sediment relocation mainly in the geomorphically most dynamic vegetation-free landscape settings (Figure 4A,B) leaving behind an exposed rocky bedrock and/or a gravel pavement surface (Figure 4E,F). Most of the country is covered by a sandy or wind-deflated rocky desert or semi-desert. Poorly developed regosolic and saline–alkaline soils prevail under a sparse desert vegetation cover and in seasonally inundated depressions, respectively. Overall poor water resources are primarily because of the scanty winter rainfall and the lack of major rivers/riverine channels flowing across/transecting this broad geographic region, most of which just with a seasonal flow discharge. Occasional summer monsoon floods add to the extremality of the Thar Desert contributing to the current environmental vulnerability of both the local natural and occupied settings.

The preserved landforms illustrate the complex geological history of the study area. The oldest exposed bedrock relics are represented by the Aravalli Hills alongside the western limits of the territory formed by the late pre-Cambrian orogeny. These are also the most ancient mountains of the Indian subcontinent and one of the oldest formations in the world [17,18]. Structural geology of the Thar Plain is built by basal igneous (granitic, rhyolithic, gneissic) rocks. The tectonic uplift followed by progressive weathering of these early magmatic bodies generated active sedimentary processes during the late Palaeozoic and Mesozoic as seen by the sequenced continental formations interstratified by marine transgressive complexes filling the deep tectonic grabens [19], sealing fossil fuel (lignite) deposits found under the desert [20]. The ancient geological units are exposed by hillslope gravity denudation forming hamadas, or wind erosion as desert pavements.

The progressing landscape restructuring during the Cainozoic era (starting ≈ 66 million years ago and continuing until now) contributed to the large scale transfer of wind-born deposits pilled up in the sand dune fields covering about 20% of the present landscape. The Thar Desert Quaternary period (the last 2.5 million years) is characterised by sequenced shifts of aeolian and fluvial processes corresponding to periodic dry and wet climatic stages, respectively [21]. The long-term tectonic and climatic processes generated cyclic accumulations of thick alluvia of palaeochannels draining the Himalaya's SW foothills [22,23]. These aquatic deposits became subsequently buried by subaerial sediments [24]. Because of the long and complex geological history, the present Thar Desert is known for numerous localities of world significance, with fossilised remains of flora and fauna embedded in geological formations that chronologically encompass a broad time range from the Palaeozoic (541–252 million years ago) and the Mesozoic (252–66 million years ago) to the Cainozoic [25]. The palaeontological (flora and fauna) records associated with the fine geological deposits/metamorphosed sedimentary rocks (mostly limestone and sandstone) provide unique evidence of the sequenced evolution of early life forms in southern Asia [26].

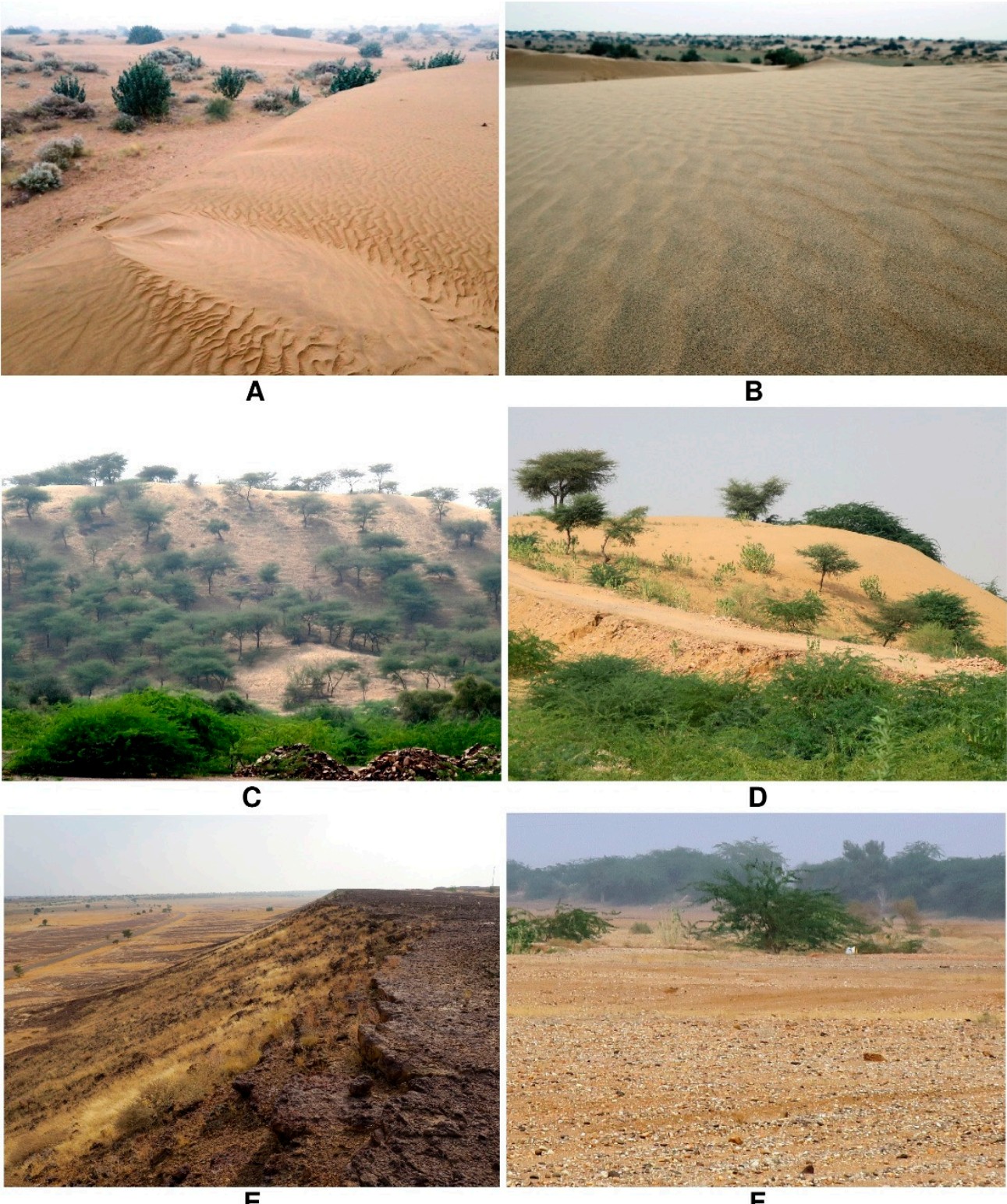

**Figure 4.** (**A**): Barchan and longitudinal sand dunes (Figure 3A) with wind-generated structural microforms, Sam (western Rajasthan); (**B**): Aeolian sand drifting of an active linear sand dune field at Kanoi (SW Thar Desert); (**C**): Parabolic dunes (30–50 m high) at Shetrawa (Figure 3B) fixed by tree plantation; (**D**): Parabolic sand dunes at Dechu stabilized by *Acacia torties*; (**E**): A wind-eroded Jurassic sandstone bedrock platform (hamada) with the surrounding rocky desert near Mokal (40 km west of Jaisalmer); (**F**): Alluvial *wadi* plain draining seasonal rainwater with wind-sorted (deflated) sandy-gravelly beds at Bhojka partly transformed into arid fields of a marginal/risky rural agriculture (ACZ I) (photographs by the author, November 2017).

### 3. Study Aims and Approaches

The present perspective on the Thar Desert environmental sustainability is in terms of the past/acting geomorphic processes, climate change and human adaptation to the wasteland habitats. It reports on the field observations and landscape documentation at the selected geosites with the contextual mapping of principal relief forms characterising the western Rajasthan. The multiproxy environmental (geographic, culture-historical and socioeconomic) assessment focused on the most vulnerable places, the ancient and modern occupation loci, and the associated natural geo-settings exposed to the ongoing aridification. Available published and unpublished database sources, and the LANDSAT images were used.

The regional environmental sustainability appraisal follows the author's field studies (2017) with an evaluation of the principal factors of the current Thar Desert landscape degradation, and the naturally and anthropogenically induced ecology risks affecting the modern settlements and land use. GIS terrain data were supplemented by satellite maps. The QGIS 3.6 program was used to construct the spatial Digital Elevation Model relief maps at the selected sites from the Shuttle Radar Topographic Mission (SRTM) database at a resolution of 1 arcsec (30 m) [27]. The projected DEM maps provided the spatial geographic and geomorphic landscape context for the issues discussed below.

### 4. Results and Discussion

#### 4.1. Desert Geo-Contexts and Modern Landscape Transformations

A broad relief mosaic characterises the present desert with the specific landforms of various scales and the corresponding local to regional/territorial geological background elements. The modern landscapes of western Rajasthan have been shaped by vigorous geomorphic processes—both endogenous and exogenous. The principal relief forms are of diverse geological geneses, including structural (orogenic ridges and volcanic dykes), denudational (hamadas, desert pavements, pediments, pediplains), fluvial (alluvial rills, gullies, terraced alluvial beds and gravelly plains of palaeo-channels), lacustrine, marine trangressive, deltaic and aeolian (sand dunes), as well as the anthropogenic landscape forms related to the past, subrecent and present rural and urban development (roads, limestone–marble–sandstone bedrock mines, industrial salinas) [10,28,29], and some geoforms with UNESCO geoheritage status [30].

Hamadas—sedimentary matrix-free Mesozoic palaeosurfaces (old planation relief relicts)—form prominent, elevated wind-abraded platforms that are exposed above the surrounding countryside (Figure 4E). These pediplain remnants [31], resulting from the long-term physical and chemical weathering with a strong ferric permineralisation [32], belong to the most authentic geomorphic features of the Thar Desert. The former depositional cover was removed by deflation exposing the pre-Cainozoic (mostly Jurassic; 201–145 million years ago) formations of hard, consolidated, coarse-sandy iron-percolated bedrock. Strong geochemical processes are quire apparent in the mineral neo-crystalline formations (inclusions) filling the original structural cavities and sedimentary nodules.

Tertiary (66–2.5 million years ago) and the earlier Quaternary riverine accumulative terraces found in the principal river valleys and preserved as erosional alluvial facies provide evidence of a rather dynamic fluvial regime with a significantly higher water volume compared to the present (Holocene) interglacial. Minor erosional gullies cutting the modern relief relate to the seasonal *wadi* channels filled by sandy gravel deposits. A considerably larger riverine catchment area compared to the present-day is documented by shallow alluvial beds in the broader valleys exposed as wind-deflated gravel pavements (Figure 4F). The fluvial gravels and weathered ferruginous nodules with desert polish together with aeolian abrasion provide witness to changing palaeoenvironmental conditions and to an early age of these ancient river accumulations subjected to strong weathering and a partial redeposition. The petrographic and mineralogical varieties of the enclosed clastic sedimentary rocks (i.e., quartz, quartzite, jasper, sandstone, marble, rhyolite) indicate their specific contextual provenance and complex geological history.

Playas—evaporate (saline) depressions—are found in the old low-relief sedimentary basins filled by clayey-silty-sandy deposits rich in gypsum and halite. The stagnant aquatic settings with a minor storage capacity and a limited drainage retain seasonal monsoonal rainwater exposed to evaporation during dry periods. The artificially water-filled salt-extraction sites (salinas) represent a relevant sector of local rural industry.

The most outstanding relief forms of the present Thar Desert are sand dune fields of diverse types, mainly linear, longitudinal, transverse, parabolic and crest-shaped barchans [33,34]. The 15–45 m high barchans are the most impressive mobile sand accumulations in the western part of the desert. Major, partly fossilised, sand dunes are found at Sam, Shetrawa and Dechu (Figure 4C,D). The ongoing spatial sand mobility co-acts with the regional desertification [35]. The general model of the Quaternary/Holocene landscape development illustrates a periodic aeolian sediment transfer leading to the sandy dunes' formation during dry periods and the landscape stabilisation at stages of increased air humidity and intensified pedogenesis under a more stable parkland vegetation cover [36]. Overall, the corroborating present trend of the continuing aridification is expressively observable in the arid zone of West India.

### 4.2. The Desert—Past Human Occupation Interactions

The present-day topography of the Thar Desert shows a dynamic geoecological and palaeoenvironmental evolution in diverse geomorphic settings and under extreme past climatic conditions. The most eloquent landscape forms—the Mesozoic hamadas and the Holocene sand dunes fields, documenting the intense erosional and accumulation processes, respectively, provide the regional geo-context of the past and modern Thar Desert occupancies. Following the orogenic stabilisation of the western part of the Indian subcontinent, the periodic atmospheric shifts played the key role in the regional relief restructuring and the pre-modern modes of human adaptation.

Numerous (geo)archaeological records and historical monuments exposed from the aeolian and alluvial deposits provide a witness of a long habitation history of this geographically marginal area. The early peoples' inhabitation of the ancient geo-settings is evidenced by Palaeolithic sites that were found contextually exposed on the ancient alluvial plains (Figure 4F) along the former palaeochannels or embedded in the stratified Early/Middle Pleistocene formations [37–39]. Wind-polished and iron-mineral tainted stone tools with the accompanying lithic waste from preparation of the prehistoric implements indicative of an early human culture occur discarded in patterned occurrences at wind-deflated gravel-pavement loci (Bojka I–II, Agolai, among other discovered sites; 2017), suggesting very favourable ancient occupation environments. The Old Stone Age artefacts of a diverse formal oblique (in view of the technological processing and surficial corrosion) and lithologies (jasper, quartz, rhyolite, basalt, etc.) provide evidence of a sequenced pre-Holocene (>12,000 year BP) peopling of the Thar basin during the presumably more humid stages of the Pleistocene period (2.5–0.012 million years ago).

The human adjustment to the (semi-)desert habitats during the following millennia is manifested by plentiful prehistoric sites and archaeological records providing a vivid testimony of an evolutionary symbiosis of people and the desert [40]. The large-scale regional erosion and sand-drifting, generating the sedimentary-matrix-free gravelly surfaces, show the intensity of the past as well as present aeolian processes [41].The former arid landscape instability and geoenvironmental shifts with the mass sediment transfers are eloquently documented by buried prehistoric localities and abandoned historical settlements that were unable to cope, at the time of their existence, with the deteriorating climatic conditions under active sand dune expansions. Periodic extreme droughts and aridification stages are apparent by the abandoned rural sites, in places re-occupied by pastoral tribes adapted to the biotically poor rocky/sandy desert habitats. The extensive animal herding has remained the principal rural economy for millennia (Figure 5C,D), with the nomadic peoples living in simple semi-permanent houses and grass-roof huts adjusted to the hot and dry climate, as they did centuries ago (Figure 5E,F).

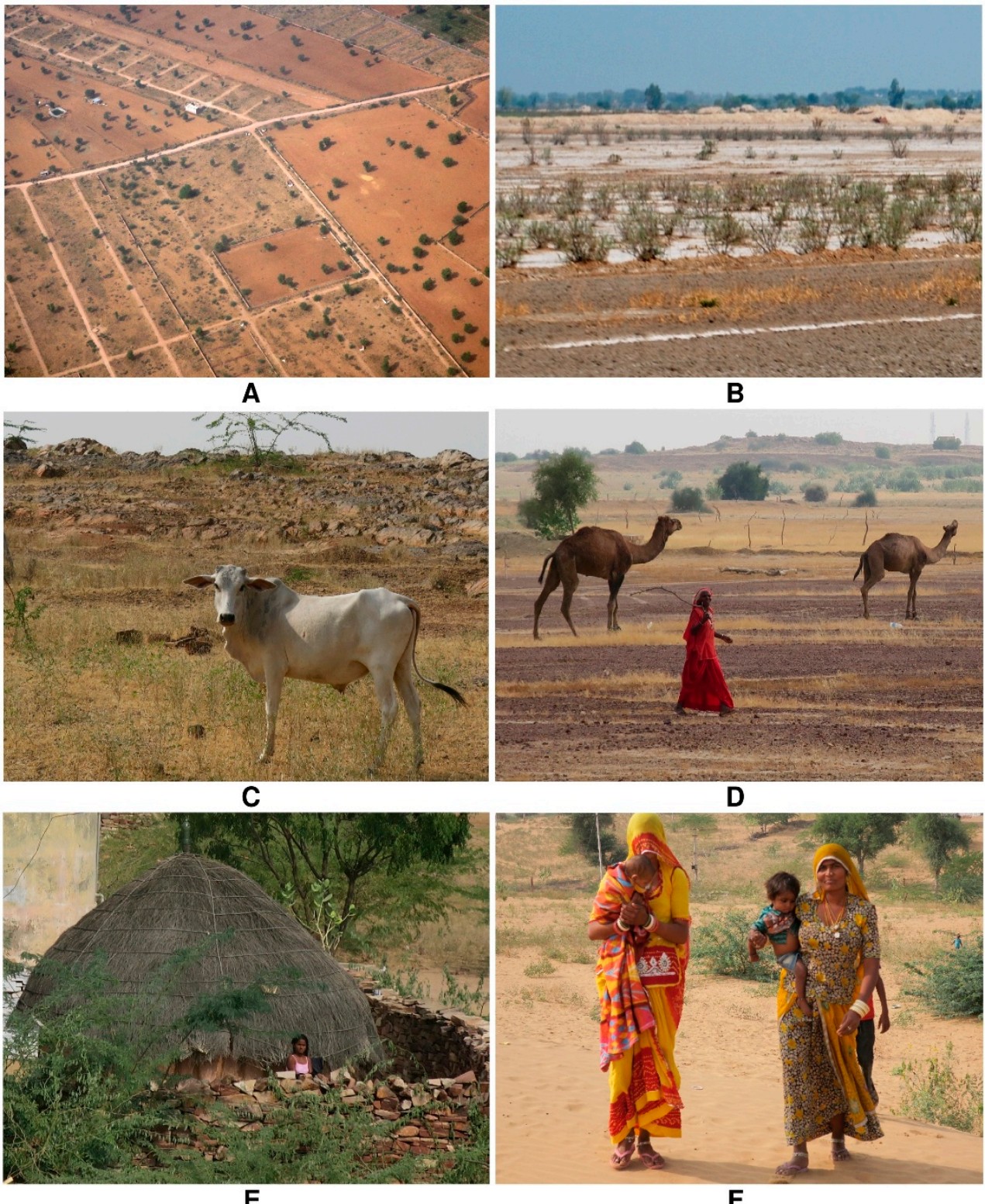

**Figure 5.** (**A**): Aerial view of the partly cultivated Thar Desert near Jodhpur with scanty bushy vegetation cover characteristic of the arid zone of NW India (AZ II); (**B**): Degraded agricultural lands (AZ IV) with top ground salinization caused by excessive irrigation and seasonal water evaporation (Pokaran); (**C**): Exposed Proterozoic (rhyolite) bedrock landscape subjected to present desertification near Agolai with perennial, seasonally dry stream channels and semi-desert vegetation used for the pastoral economy; (**D**): Arid rocky desert land near Mokal used for camel herding; (**E**): Traditional grass-roof huts of the Thar rural inhabitants with a stone-slab circular alignment; (**F**): Women of the Gadia Lohars nomadic tribe, the Dechu sand dune field, western Rajasthan (photographs by the author, November 2017).

The past natural conditions, along with the regionally and locally specific landscape evolution, predisposed the arid land ancient/early historical settlement sustainability. The existence of hydrological bodies with year-round drinking water accessibility was the key occupation factor as it is today. In turn, the aquatic-resource availability extent pre-determined the forms of the early (hunting-gathering/pastoral/agrarian) economies.

### 4.3. Present Climate Change Geoenvironmental Response

The present climate development in western Rajasthan—the eastern Thar Desert area—is evidenced by (1) the progressing landscape aridification despite certain raising trends in annual precipitation at some locations; (2) the strengthened regional SW-NE oriented windiness accompanied by the intensified mass aeolian-sand transfer into cultivated lands (Figure 3); (3) the activation of sand dunes because of sparse grassy vegetation; 4) the shifts in the regional hydrology with the decrease in the seasonal runoff in the local streams (*wadis*). Western Rajasthan experiences the highest long-term deficit of annual and seasonal precipitation on the Indian subcontinent with a highly negative climate development forecast [42] predicting further, more spatially extensive desertification in the coming years fundamentally affecting the local ecosystems.

The geomorphic aspects of the areas subjected to desertification play the most important role in the current landscape transformations [43]. The desert relief configuration provides evidence of large-scale surface-cover changes due to the co-acting erosional and sedimentary actions generating the wind-deflated rocky landscapes and the active sand dune fields, respectively [33,44], with the insolation increased by the barren land exposure. The dynamic aeolian activity with sand drifting and the resulting bed-forms are particularly evident in places of sparse vegetation and a low mean annual rainfall (<200 mm/year). These forms are most-expressed in the longitudinal/barchanoid and linear dune fields near Sam and Kanoi, respectively, and by the partly stabilised and up to 30 m-high parabolic dunes near Dechu and Shetrawa, among other desert places (Figure 3A,B and Figure 4A,B). The Sam dune field, ≈45 km to the west of Jaisalmer, forms a chain of prominent barchans—the major of its kind in the Thar Desert. Wind-deflation depressions between the dunes illustrate the vigour of these aeolian processes with the sand transport onto the pastoral and agricultural lands over a large (tens to hundreds of kilometres) distance.

The sedimentary matrix-striped grounds at the mapped sites of Chandan and Bojka with the exposed Pleistocene gravelly pavements attest to the intensity of wind erosion (Figure 4F). These alluvial facies, which were a partial source of the sands amassed in the dunes, are stratigraphically underlain by a moderate concretionary lime zone as a result of the past sediment weathering and subsequent pedogenesis, suggesting a more humid (pluvial) climate than today. The present-day warming also contributed to an expansion of the shallow saline depressions (playas) interlinked by perennial streams seen at Lawan, Pokaran and Khara Bhagotiya, among other places [45], and the seasonally filled natural ponds amidst the open sandy terrain. These evaporate places rich in dissolved salt minerals are subjected to high evaporation (1800–2000 mm) during the dry periods following the summer monsoon rainfalls. The pre-monsoon season in the area documents the current climate pattern of significant warming during the spring months (April–May), with the mean decadal daily temperature rise by ≈1 °C—the most pronounced over India [46]. Finally, the adverse climate effects to the geoenvironment also mean a lower rate of vegetation growth and a slower (regosolic) soil cover restoration.

The present climate impacted the desert landscapes, both occupied and barren, also shows accelerated ground-surface erosion by the seasonal floods due to the low surface-cover water retention followed by an activated mass sheet aeolian transfer The arid-zone (semi-)desert lands are thus exposed to the large-scale sand mobility and loose sediment burial, as well as fluvial erosion, both representing the major geo-hazards with respect to the future countryside exploitation and settlement potential. Weather storminess may further generate a massive (up to >1 t/ha) soil loss during a single event [11], causing a progressive land degradation and natural nutrient loss over a short period.

### 4.4. Desertification Risks and Environmental Mitigation

Desertification, groundwater salinity and soil nutrient loss are the common ecology risks in the desert regions of western Rajasthan. The environmental countermeasures to the continuing territorial aridification include the construction of facilities for the irrigated agriculture-based water management in the form of artificial discharge channels and rainwater pond storage systems (*khadins*). Excessive irrigation from new artesian well drillings causes groundwater depletion, ground waterlogging and salinity of the cultivated lands leading to their decreasing productivity. A partial solution is saline water (re)utilisation practiced in other desert regions of south/central Asia and Africa [47].

Groundwater management is also a lasting strategic issue in the western (Pakistan) part of the Thar Desert [48]. The water absorption and recharge capability of the desert sand dunes were suggested as potential water retention sources [49]. The recharge rates under rain-fed agriculture (with >300 mm/year precipitation) with irrigation of 20–40% of the overall cultivated land is presumed as sustainable [49]. Yet, in view of the high evapotranspiration rate and the high permeability of sandy deposits the relevance of the seasonal dune intake of water for the regional annual hydrogeology balance is relatively minor. The dune water-absorption mechanism may have local significance for the nomadic rural settlements along with artificial watershed development [50].

Arid-zone agroforestry plays a crucial land-management role that is aimed at the crops' production, as well as the wasteland mitigation [51,52]. The agrarian land protection programme launched by the Government of Rajasthan in 1978 led to a partial dune stabilisation and wind-blown sand fixation by afforestation of the dunes' levee slopes, which is well exemplified at Shetrawa and Dechu (Figure 4C,D). The xerothermic, dryness-resistant tree plantation contributes to the wasteland wind erosion control, the sand dune demobilisation, the artificial watershed and rainwater retention, the adaptive desert land farming, arable soil protection, ground desalinisation, and degraded land recovery, and the overall environmental remediation and restoration of western Rajasthan.

Protective ecological responses against desertification with the implementation of novel and relatively low-cost biotechnologies balance to some extent sustainability of the state's future socioeconomic development. Monitoring of seasonal flood effects to relief, experimental studies with simulations of the local hydrology system behaviour and the monsoonal rains' gravity slope-wash [53], as well as the modelling of the climate impact contribute to natural hazard resiliency assessments. Other desert landscape protection issues include a geomorphic mapping and landforms' analyses in relation to the land-use practices generating some forms of anthropogenic environmental degradation [13,54–56].

Analogous settlement ecology problems related to the current climate variations and the associated geoenvironmental shifts and transformations due to desertification principally affecting rural areas are encountered in other arid regions worldwide [57–59]. These are responded to by the locally specific natural sustainability policy measures and development actions including the water-resource (re)utilisation, degraded/threatened landscape restoration and wasteland reforestation [60–68]. The Thar Desert ecosystem mitigation approaches follow in part the pre-modern water- and soil use practices and strategies that are documented in other desert areas of central and southern Asia [69–71], as well as the traditional Indian water-management systems [72].

Finally, the desert's historical and, in particular, modern settlements have locally significantly distorted the natural relief via large-scale constructions and intensified agrarian activities. Human actions and land-management practices generated new biophysical factors that contribute to land degradation [73]. A differential anthropogenic impact is particularly evident in the formerly pristine marginal areas of the Thar basin mainly due to the intensified irrigation agriculture and rural pastoralism, as well as the raw-material resource extraction, which all require infrastructure (roads) building across the desert. Rock and mineral mining (feldspar, phosphorite, gypsum, kaolin, limestone and marble among other deposits) is also a massively expanding industry. These economic activities linked to the continuing peopling of the formerly barren lands, along with climate change, pose

immediate threats to the pristine desert environments, which fundamentally affect the traditional lifestyles of nomadic people.

*4.5. The Thar Desert Sustainability Management*

The Thar Desert has the highest density of population among the world's arid zones. Rural agriculture and pastoralism are the principal traditional economies of the wastelands. Four agroclimatic zones (ACZs I–IV), representing the environmentally and geographically defined land sustainability units characterized by specific physiographic features and socioeconomic factors, are found in Rajasthan [10,13]. Zone I includes extensive and environmentally marginal lands with risky agriculture, which depends entirely on a scanty annual rainfall (150–250 mm/year) reduced by high evapotranspiration. About 60% of this area is covered by various (barchanoid, parabolic, linear, transverse) sand dunes, up to 50 m high (Figure 4A–D). Desert vegetation is sparse with the dominance of xerophytic grasses and shrubs. Zone II (<300 mm annual rainfall) encompasses two agriculturally prosperous districts (Sri Ganganagar and Hanumangarh), where the original dunes' terrain was transformed into relatively stable flat plains after surface levelling and ploughing for seasonally irrigated fields (Figure 5A). Zones III and IV, constituting transitional plains with internal drainages, are located in the more densely populated eastern, sand dune-free districts of Rajasthan, which receive an increased annual rainfall (300–500 mm) and a significant seasonal riverine discharge that delivers nutrient-rich sediments. However, excessive watering of the cultivated fields leads to top-ground salinity and permanent land degradation (Figure 5B). Soil cover protection and preservation are the key goals of the regional agrarian policy [74].

Despite the continuing territorial aridification, the rain-fed croplands (mainly millet) have locally extended at the expense of the traditional pastoralism in all the agroclimatic zones responding to the rising demography. More than 51% of the kept livestock became concentrated in the western, most arid part of the state (the Thar Desert), where herding (≈21% cattle, 13% buffalo, 23% sheep, 41% goat, and 2% camel) has been the key livelihood base of the local nomads for centuries (Figure 5C,D). The current trend in the reduction of grazing lands in most of the Rajasthan districts indicates (1) the negative environmental impact of livestock pressure to the sparsely-vegetated landscapes; (2) the intensified plantation land-use; and (3) the changing climate regime (aridification). The balanced traditional forms of the rural economy along with rangelands fencing and controlled grazing enhanced the local xerophytic biotope resilience to environmental stress. Positive trends in population density and distribution of khejri trees (*Prosopis cineraia*) used by the desert dwellers is ascribed to a certain rainfall increase (long-term observed in the eastern and NW parts of Rajasthan) [75] and agroforestry management [76].

The present natural sustainability of the Thar Desert occupation habitats is pursued by the implementation of alternative land-management approaches and new agrarian technologies within the steadily extending (semi-)desert countryside. The mapping and monitoring of the sand dune expansion and the vegetation cover stability, assessment of the landscape feedback to current climate change, the protection of the productive plains against desertification, the desert farmland yield enhancement as well the pristine nature preservation and biodiversity protection are the key environmental priorities [8]. The physical fixation of the loose aeolian sedimentary cover and the water-resource management are the leading strategies against the progressing regional aridification and the desert enlargement beyond its present geographical limits.

The integrated geoecology investigations detailed the transformations of the past and present Thar Desert relief related to desertification, as well as the natural landscape alterations/distortions due to modern human actions and the increasing anthropogenic pressure on the pristine environment. Complex applications of the novel research results and the new environmental knowledge implementation is essential for the successful mitigation of the degraded and threatened desert lands by taking into account the water availability differentiation due to the climate variability (with a recurrence/increasing

frequency of drought episodes), the land topographic diversity, the local hydrogeology conditions and the dominant plant-cover water demands.

The arid-zone settings predisposed by the specific geographic and geomorphic configuration and the atmospheric regime gave rise to a variety of biotopes and ecosystems characteristic of the territorial climatic and geographic zonality [77]. The original natural settings are becoming more vulnerable to the climate- and human-generated changes. The Thar National Park (a UNESCO World Natural Heritage Site)—the largest (3162 km$^2$) among the biosphere reserves and the protected natural areas of western India—includes, except for scenic landscapes, pristine desert habitats with unique wildlife [78]. Biogeography preservation of the most threatened places is of utmost importance.

Travel and exploration increasingly contribute to the present development of West India's arid zone. As one of the culturally most colourful and authentic states of India, Rajasthan has a major potential for geo/ecotourism and recreation, which may further contribute to the geoenvironmental stability of the country and the continuing socioeconomic development with livelihood advancement of the rural desert areas.

The long-term nature sustainability priorities of the arid western Rajasthan include:

- land recovery/protection measures against water shortage and land-erosion risks;
- environmental remediation and regional ecology stability with balanced land use (including innovative crop cultivation, agriculture diversification, traditional pastoral economy avoiding grasslands' overgrazing and artificial plantation);
- socioeconomic awareness aimed at the sustainable exploitation of natural resources;
- protection of biotically the most significant geoecosystems with endemic flora and fauna, along with the current and predicted biodiversity threats' monitoring;
- geo/ecotourism and geoheritage promotion of the Thar Desert area rich in unique palaeontological sites, geosites, cultural monuments and historical places [79–82].

## 5. Conclusions

The Thar Desert constitutes the most geomorphically dynamic area of the Indian subcontinent. The modern landforms' diversity with the prominent planation (hamadas) relief and the extensive sand dune fields illustrates the complex regional geological history. The synergic effects of the increasing MAAT and the negative annual precipitation trend, which intensifies the ground surface evapotranspiration and wind-erosion processes, are the main climate-change generated constraints. The current territorial aridification, as evidenced by the mass aeolian sediment transport into the cultivated lands and the groundwater level drop, leads to settlement instability. The ongoing desert expansion represents an adverse factor with respect to the economic development, the ecological integrity, and environmental sustainability of western Rajasthan. Landscape aridification is partly balanced by the agrarian remediation through selected xerophytic crop cultivation and a return to the traditional pastoral economy. In view of the large-scale impulses of climate warming, the combating ecology strategies only have a limited effect over the long-term period. Studies of the contemporary atmospheric variations affecting the present desert's relief and geo/biodiversity contribute to a better understanding of the ongoing environmental transformations of this most arid territory of NW India. The countless prehistoric localities and the preserved, sand-exposed archaeological monuments provide a vivid testimony of the long and rich cultural history of the Thar Desert and the peoples' vital adaptations to the extreme desert settings and the biotically marginal natural habitats.

**Funding:** The research received external funding from the Adam Mickiewicz University in Poznan and the Environmental Research Centre in Stare Mesto.

**Institutional Review Board Statement:** Not applicable.

**Informed Consent Statement:** Not applicable.

**Data Availability Statement:** Data supporting the reported results can be found at the IGG AMU in Poznan, Poland, and the Environmental Research Centre in Stare Mesto, Czech Republic.

**Acknowledgments:** The geomorphology and environmental study in the Thar Desert area of western Rajasthan was carried out with a support of the Institute of Geoecology and Geoinformation, AMU, Poznan. Pratap C. Moharana (the CAZRI, Jodhpur) kindly commented on a draft of this paper and provided additional information on the present land use of the discussed area. Jolanta Czerniawska (Institute of Geoecology and Geoinformation, AMU, Poznan) prepared the DEM maps (Figure 3).

**Conflicts of Interest:** The author declares no conflict of interest.

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
