# Peer review of "Between Sand Dunes and Hamadas: Environmental Sustainability of the Thar Desert, West India"

_sustainability, doi:10.3390/su13073602_

Round 1

Reviewer 1 Report

The article develops a topical issue - environmental sustainability.
There are several points of the article that should be improved:

  • For the figure 1 add the source (own processing).
  • Is the scale of the map the same for both figures in Figure 1? if not, add the scale to the map on the top right of figure 1.
  • Put the acronym MAAT after you first wrote the full name.
  • In figure 2 mention the source for figures A and B (own processing). 
  • To the legend of figure 2 add the sign for the settlement.
  • In figure 3, point A, after the “digital elevation model” put in brackets the acronym (DEM). This acronym is found at point B.
  • There are no references in the text for figure 4 (C, D).
  • Put the source at figure 4. For example: personal archive, November 2017.
  • Mention the resolution of LANDSAT satellite images.
  • Mention for the archive maps which is the institution that provides them.
  • Add a sketch/diagram of the geological time scale to the materials and methods. This diagram is required for readers who do not have information about geological time.
  • At the the row 214 change the word ”fiure” to ”figure”.
  • Add in the text the reference for figure 5 (A, B).
  • Detailed what mean AZ II and AZ IV. It can not be understood...
  • I recommend you add to the source "photographs by the author" of the figure 5 the month and year when they were taken.
  • At the point 4.3. I recommend you to add a short description about the climatic context of the study area. From this description to show observed changes for temperature, precipitation, evaporation wind speed.
  • At the the row 344 change the word ”fiure” to ”figure”.
  • At the rows 376 and 377 you need to use the same font in editing. 
  • I recommend you to enter a table where to centralize the information about the agro-climatic zones described in the text of the point 5. It is difficult to follow...

Reviewer 2 Report

Dear Author,

I read your manuscript with great interest. It is nicely written and well illustrated.

However, I do not feel that this can be considered as a research 'Article'. Apparently, the manuscript does not originate from a original research, and in its present form it would better suit as book chapter rather than a journal article. The classical titles of the sections do not match with their content. In fact, the very brief section on 'Material and methods' hardly reports which are the data on which the research is based and the methods used.

The same applies to the 'Results', which apparently do not report outputs originating from an original research.

I see two options. Either the manuscript is converted in a 'Review' paper enlarging the spectrum of bibliographic background or it is re-submitted after a thorough revision which makes clearer to the readers which are the aims of the research and based on which methods and tools it has been conducted.

In the annotated manuscript a series of suggestions and comments are provided. 

Regarding more general aspects, please consider what follows:

  • Indicate the aims of the paper in the Abstract
  • Better related the titles of sections 3 and 4 to their content
  • Clarify the content of lines 151-155 (specific comments are provided in the annotated manuscript)
  • Reduce size of Figures 1-2-3
  • Use the journal style for citations in the text.

Round 2

Reviewer 1 Report

Dear Author,

I appreciate that you have taken into account my suggestions to improve your article. Thanks!

But:

  • A development / improvement of section 3 (Study aims and approaches) is needed; 
  • Reformulate the first sentence of section 3. 
  • Describe your ”personal field studies” in section 3.
  • Please document yourself for the correct spelling of the term geosite and clarify the use of the terms paleontological sites and geosite in your text.

All the best!

Author Response

Dear Author,

I appreciate that you have taken into account my suggestions to improve your article. Thanks!

  • A development / improvement of section 3 (Study aims and approaches) is needed; 
  • Reformulate the first sentence of section 3. 

Done.

  • Describe your ”personal field studies” in section 3.

provided

  • Please document yourself for the correct spelling of the term geosite and clarify the use of the terms paleontological sites and geosite in your text.

(both terms are generálky known; added ref. 81-83 „geo-site“ definition)

All the formerly requested (Review 1) improvements and specification in Section 3 were largely provided, including the aims of the study and the personal (=author’s) work (stated in the text, 2nd paragraph of the section). Any additional comments would be redundant and will distort the already formatted text. The study aims are also clearly stated (p. 3, 3rd paragraph)

Both forms geo-sites and geosites are OK, unless are referred to consistently in a text.

The English language and style checked once again with some corrections.

Reviewer 2 Report

Dear Author,

thanks for taking into account my comments and suggestions. Now the structure of the manuscript is more effective. Yet, I believe that Section 3 (Study Aims and Approaches) should be improved (see comments below).

Lines 166-170: The new sentence is not very clear and too long. Its content should be better clarified. Splitting the sentence into two parts would probably help.

Lines 170-171: It is not explained how the 'multi-proxy environmental (geographic, culture-historical and socio-economic) assessment' has been carried out. 

Line: 174: 'Personal field studies' is very vague. What type of studies?

Regarding the rest of the text, please not what follows.

Line 27: 'world-wide' should read 'worldwide' as in other parts of the manuscript.

Lines 169 and 474: 'geo-sites' should read as 'geosites' according to the main literature on the topic.

Line 474: Do you need to distinguish 'geosites and 'palaeontological sites'?Normally, site of paleontological interest are comprised within the general term 'geosites'. Or do do you mean something different?

Thanks for your attention.

Author Response

Dear Author,

thanks for taking into account my comments and suggestions. Now the structure of the manuscript is more effective. Yet, I believe that Section 3 (Study Aims and Approaches) should be improved (see comments below).

Commented on in the Review 1.

Lines 166-170: The new sentence is not very clear and too long. Its content should be better clarified. Splitting the sentence into two parts would probably help.

Done.

Lines 170-171: It is not explained how the 'multi-proxy environmental (geographic, culture-historical and socio-economic) assessment' has been carried out. 

Review assessment.

Line: 174: 'Personal field studies' is very vague. What type of studies?

Geography, geomorphology, geoarchaeology, environmental (lines 170-171) .

Regarding the rest of the text, please not what follows.

Line 27: 'world-wide' should read 'worldwide' as in other parts of the manuscript.

corrected

Lines 169 and 474: 'geo-sites' should read as 'geosites' according to the main literature on the topic.

Both ways of spelling are correct.

Line 474: Do you need to distinguish 'geosites and 'palaeontological sites'?Normally, site of paleontological interest are comprised within the general term 'geosites'. Or do do you mean something different?  

A specific geo-site with palaeontological occurrences as opposite to a geo-site as a geologically and/or geomorphically interesting place without fossil findings. For clarification, additional references provided.

Thanks for your attention.

Thank you to your comments and constructive suggestions.